# The Celestial Frame and the Weighting of the Celestial Pole Offsets in the Computation of VLBI-Based Corrections for the Main Lunisolar Nutation Terms

**DOI:** 10.3390/s21248276

**Published:** 2021-12-10

**Authors:** Víctor Puente, Marta Folgueira

**Affiliations:** 1National Geographic Institute of Spain, General Ibañez de Ibero 3, E-28003 Madrid, Spain; 2Department Section of Astronomy and Geodesy, Faculty of Mathematics, University Complutense of Madrid, E-28040 Madrid, Spain; marta_folgueira@mat.ucm.es

**Keywords:** nutation, celestial pole offsets, VLBI

## Abstract

Very long baseline interferometry (VLBI) is the only technique in space geodesy that can determine directly the celestial pole offsets (CPO). In this paper, we make use of the CPO derived from global VLBI solutions to estimate empirical corrections to the main lunisolar nutation terms included in the IAU 2006/2000A precession–nutation model. In particular, we pay attention to two factors that affect the estimation of such corrections: the celestial reference frame used in the production of the global VLBI solutions and the stochastic model employed in the least-squares adjustment of the corrections. In both cases, we have found that the choice of these aspects has an effect of a few μas in the estimated corrections.

## 1. Introduction

The study of the nutation of the Earth is fundamental in two aspects: to infer the structure of the Earth’s interior and to determine the orientation of the Earth’s axis in the inertial frame. Very long baseline interferometry (VLBI) is the technique that allows the most regular and precise observations of the nutation of the Earth by means of the observations of extragalactic radio sources, although according to [1], lunar laser ranging (LLR) has also the potential to determine celestial pole offsets (CPOs) with an accuracy comparable to VLBI. The CPOs represent the mismatch between the IAU 2006/2000A precession–nutation model and the observations.

Several analysis centers (ACs) contribute to the International VLBI Service for Geodesy and Astrometry (IVS, [2]), providing EOP time series estimated from the analysis of VLBI observations. The IVS combination center combines the estimates from the different ACs to produce the combined solution [3]. Therefore, time series of CPOs can be extracted both from an individual AC’s solution and from the IVS combined solution. The IVS series of CPOs are later processed and compiled in the IERS long-term series [4] with the rest of the EOPs that are produced by means of other space geodesy techniques under the umbrella of the corresponding services (the International GNSS Service, IGS; the International Laser Ranging Service, ILRS; and the International DORIS Service, IDS). 

Using the available series of CPOs as a starting point, it is possible to adjust amplitudes to several terms of the IAU 2006/2000A precession–nutation theory in order to identify signals that are not included in the model. In this sense, Ref. [5] used the CPO solutions from nine IVS ACs and three combinations to estimate corrections to the IAU 2006/2000A precession–nutation model. Additionally, they estimated supplementary terms accounting for the precession rate, the misalignment of the celestial reference frame and a retrograde circular term to subtract the effect of the free core nutation (FCN).

On the other hand, Ref. [6] studied the impact of frames, and other processing strategies, on the CPO estimates, using the resulting series of these tests to readjust the precession offset and rate and the main nutation amplitudes available in the IAU 2006/2000A model. More recently, Ref. [7] estimated corrections to the main nutation terms within the VLBI analysis process (direct approach) and compared the results to the ones obtained using CPO series (indirect approach). They found that the direct approach returned lower uncertainties, lower correlations between nutation correction estimates and, in some cases, significantly different results with respect to the indirect approach. Another contribution to this research topic is the work by [8], which estimated corrections of lunisolar nutation components from six sets of CPOs and used the remaining residuals for the study of the FCN signal.

The usual approach in the estimation of corrections to the main lunisolar nutation components is to take as input the latest available CPO series at the time, without taking into account their compatibility in terms of analysis configuration. This point was already noted by the authors of [5] when they found significant differences in the corrections derived from each solution. These differences were due to the analysis configuration or the software packages used by each AC. The authors highlighted the need to research the reasons that could explain the differences between the series. In this work, we assess the impact on the corrections to the main lunisolar nutation terms of one relevant factor in the configuration of VLBI analysis, which is the celestial reference frame. We complement this analysis with an assessment on the weighting of the CPO in the least-squares adjustment of the nutation corrections, following an approach that is different from [5]. This new approach is based on scaling the uncertainties of the CPO as a function of the network geometry of each VLBI session used in the analysis.

## 2. Methods

For the computation of the corrections to the IAU 2000A model, we adopted the lunisolar nutation terms used by the authors of [9] in their work. These terms consisted of 21 prograde and retrograde circular terms with known astronomical frequencies and phases. The corresponding harmonic terms and periods are listed in Table 1. Columns 2 to 6 correspond to the multiplier factor of the Delaunay arguments [10] and are detailed below:l: mean anomaly of the Moon,l’: mean anomaly of the Sun,F: mean longitude of the Moon minus mean longitude of the ascending node of the Moon,D: mean elongation of the Moon from the Sun,Ω: mean longitude of the ascending node of the Moon.

The mathematical model used for the computation of corrections to the main nutation terms of the IAU2000A model by means of a least-square harmonic fit is shown in Equation (1):(1)$$\begin{array}{c} {dX\text{-}\mathrm{dX}}_{\mathrm{FCN}}=\sum_{i=1}^{42}\left[a_{c,i}\cos\left({\mathrm{ARG}}_{i}\right){-a}_{s,i}\sin\left({\mathrm{ARG}}_{i}\right)\right],\\ {dY\text{-}\mathrm{dY}}_{\mathrm{FCN}}=\sum_{i=1}^{42}\left[a_{c,i}\sin\left({\mathrm{ARG}}_{i}\right){+a}_{s,i}\cos\left({\mathrm{ARG}}_{i}\right)\right], \end{array}$$
where (*dX*, *dY*) are the CPOs available from the VLBI series, (*dX_FCN_*, *dY_FCN_*) account for the effect of free core nutation, (*a_c,i_*, *a_s,i_*) are the amplitudes of the corrections and *ARG_i_* are linear combinations of the fundamental arguments of the lunisolar nutation theory [10] listed in Table 1. In all the tests presented in the following sections, the corrections to the terms listed in Table 1 were computed after having removed the FCN signal using the B16 model [11]. 

## 3. Results

This section includes the results of the two tests performed for the computation of corrections to the IAU 2006/2000A precession–nutation model.

### 3.1. Test A: Different Celestial Reference Frame

The first test was to estimate the corrections using as input the series compatible with ICRF2 ([12]) and ICRF3 ([13]) for each of the AC considered in Table 2. Only those IVS ACs providing global solutions with both ICRF2 and ICRF3 were considered. The solutions for each AC were selected following the criteria of using the last available series compatible with ICRF2 and the first series obtained using ICRF3. The metadata of the solutions were reviewed to ensure that there were no other major changes in the configuration of the analysis; this was also the reason to use several solutions, in order to mitigate the potential differences in the processing between the two solutions of the same AC. The time span of the analysis was limited to 1993–2021, given that data before 1993 presented worse accuracy and lower temporal resolution than more recent data [11]. 

Figure 1 and Figure 2 show for each solution the differences between the corrections derived from the ICRF3 and ICRF2 series, for *a_c,i_* and *a_s,i_* respectively.

In general, the differences were in a band of ±5 μas for all the periods, except for the ASI solution which showed a more scattered behaviour. Table 3 includes the range of the corrections for each AC together with the mean formal error of the residuals in brackets. It also includes the solution (ICRF2 and ICRF3) and the median and standard deviation (STD) of the differences between the corrections estimated with the series associated to each ICRF.

The median values of the corrections presented a similar behaviour regardless of the ICRF used. The median value of the range (i.e., the difference between the highest correction and the smallest correction with their sign for the 42 terms) for the *a_c,i_* amplitudes was 22.3 μas for ICRF2 and 26.1 μas for ICRF3. For the *a_s,i_* amplitudes, the median value was 26.7 μas for ICRF2 and 28.0 μas for ICRF3. The analysis of the range of the corrections was aimed at noting the magnitude of the estimated corrections and also at pointing out the consistency between the different AC solutions. 

The differences between the corrections had a median value of −0.1 μas and standard deviation of 2.0 μas for *a_c,i_* amplitudes and a median of 0.4 μas with standard deviation of 1.7 μas for *a_s,i_* amplitudes. Finally, it should be noted that the use of ICRF3 did not improve the formal error of each computed correction. The mean formal error over all the AC solutions was 2.2 μas for both ICRF2 and ICRF3. In addition, no significant difference was found between the solutions of the different ACs.

From these results it can be concluded that the difference between using ICRF2 or ICRF3 was in the level of a few μas (band of ±5 μas) whereas the estimated values of lunisolar nutation corrections were in the order of a few tens of μas. Therefore, the choice of ICRF for the estimation of corrections to the nutation model had an impact of one order of magnitude less than the magnitude of such corrections.

### 3.2. Test B: Different CPO Weigthing

In the results presented in the previous section, no stochastic model was used in the least-squares adjustment to estimate the corrections to the nutation model, whereas [5,6] used weights taken as the inverse of the squared formal errors from the VLBI series. Additionally, Ref. [5] estimated a scale factor and an error floor together with the fitting of the corrections in order to account for the differences in the observation geometry between VLBI sessions.

One of the main factors that impacts VLBI estimates of EOPs is the network geometry, and this fact can be taken into consideration in the estimation of corrections to the IAU 2000A model from VLBI-based CPO series [14]. For this purpose, the CPO precision as a function of the network of the corresponding VLBI session needs to be rescaled and this information can be used to build a stochastic model.

The relationship between network geometry and EOP precision was already addressed by [14]. Following an analysis of the VLBI series from June 1996 to February 2007, the authors found that the volume of network (*V*) could be used as an indicator of the uncertainties in EOP estimation. In order to have consistent uncertainties between sessions, they should be reduced to the unit volume of the network as follows:(2)$$\sigma_{m}=\frac{\sigma }{V^{c}}\;$$
where *σ_m_* is the modified uncertainty, *σ* is the original uncertainty, *V* is the volume of the network and *c* is a parameter that relates EOP precision and network volume. This value can be fitted using long-term series of VLBI EOP estimates, as will be detailed later.

In order to have consistent uncertainties to build an appropriate stochastic model, we have followed the same methodology as [14], extending the analyzed period to fit a new model that relates network volume and uncertainty. As a proof of concept, we used GSF series (gsf2020a.eoxy) in the period 1993–2021. The reason for using only one solution was that in this case the factor under analysis was the geometry of the observing network, which was not AC dependent. The geometry of the sessions was the same for all ACs.

The following computations were carried out based on [14]:

1. For each session of the series, the tetrahedron mesh corresponding to the network polyhedron was defined by means of the Delaunay triangulation. The list of stations in each session was extracted from the EOP series, which included the two-letter IVS station identifiers. The Delaunay triangulation was computed using MATLAB [15].

2. Computation of the volume of each tetrahedron (*v_i_*):
(3)$$v_{i}=\frac{1}{6}\left|\left({\vec{r}}_{2}-{\vec{r}}_{1}\right)\cdot \left(\left({\vec{r}}_{3}-{\vec{r}}_{1}\right)\times \left({\vec{r}}_{4}-{\vec{r}}_{1}\right)\right)\right|,$$
where *r_i_* is the geocentric station vector. Sessions that had less than four stations were not considered in the analysis.

3. Computation of the total network volume (*V*) as the sum of the volumes of all the tetrahedrons computed in the previous step.

4. The EOP series were binned in nine groups based on the total network volume in Mm^3^
$V<0.1,\;{\left(\sqrt{10}\right)}^{k}\leq V\leq {\left(\sqrt{10}\right)}^{k+1},\;k=-2,\text{…},5.$

5. For each group, the average network volume ($\bar{V}$ and average CPO uncertainties ($\bar{\sigma_{dX}}$, $\bar{\sigma_{dY}}$) were computed, obtaining the values in Table 4.

6. Least-squares fitting of a power model relating CPO precision (*σ*) and network volume (*V*) with the following form:
(4)$$\log\overset{¯}{\sigma }=b+c\cdot \log\overset{¯}{V},\;$$
where *b* and *c* are the parameters to be estimated. This fitting was applied for *dX* and *dY* separately, and the results are shown in Table 5. The value of *c* was within the same order of magnitude as the one obtained by [14] for a shorter period of analysis (−0.238). The difference in the *b* value ([14] reported −0.772 for *dX* and −0.772 for *dY*) could be due to the number of sessions considered in the analysis (1440 in [14] and 2268 in this work). 

Figure 3 shows the relationship between CPO precision and network volume. Dots correspond to the logarithmic values of Table 4 and the solid line is the fitted model. Logarithmic scales were used on both axes. This figure was similar to the one obtained by [14] for a shorter period of analysis and it also showed a clear relationship between CPO precision and network volume.

It was possible to perform a comparison of the effect of the three different stochastic models in the weighted least-squares adjustment to estimate corrections to the nutation model. The three scenarios analysed were the following:
Test B.1: no stochastic model.Test B.2: original uncertainties from the series. In this case, the elements of the weight matrix were:
(5)$$w_{i}={\left(\frac{\sigma_{0}}{\sigma_{i}}\right)}^{2},\;$$
where *σ*_0_ is the a priori variance factor and *σ_i_* is the uncertainty available in the original
series. *σ*_0_ was set to the mean value of the uncertainties in the original series.Test B.3: Modified uncertainties. In this case, the elements of the weight matrix were:(6)wi=(σ0σiVc)2, 

Two statistical criteria were used in order to compare the results of the different strategies evaluated on the CPO weighting:
A posteriori variance factor ($\hat{\sigma }$): quotient between the sum of each weighted residual square and the number of degrees of freedom of the least-squares fitting.χ^2^ value quotient between the a posteriori variance and the a priori variance factor. Values of χ^2^ close to one indicated a realistic adjustment in terms of a priori weights. 

The results obtained in the three scenarios analysed are shown in Table 6, where it can be seen that the test in which the modified uncertainties were used to build the stochastic model brought the smallest a posteriori variance factor and the χ^2^ value closest to 1. Therefore, it can be stated that this was the most suitable stochastic model to be used in the weighted least-squares adjustment to estimate corrections to the nutation model. The improvement in the formal error of the corrections is below μas when using this approach.

Finally, Table 7 includes the statistics of the differences in the corrections considering the best option (Test B.3) with respect the other two tests performed on the CPO weighting. From these results, it can be concluded that the choice of the stochastic model produced significant differences in the corrections since they were in the μas level, which was the level of the accuracy of the IAU 2000A nutation model

The corrections computed in Test B.3 (modified uncertainties) presented median differences with respect to the other two tests in level of 1 μas or below, standard deviation between 4–5 μas and a range in differences of a few tens of μas. This range was computed as the difference between the highest correction difference and the smallest correction difference with their sign.

## 4. Conclusions

In this paper we studied the influence of the celestial reference frame and the stochastic model of the least-squares fitting in the estimation of empirical corrections to the main lunisolar nutation terms of the IAU 2006/2000A precession–nutation model. As a starting point, we used CPOs derived from global VLBI solutions. 

For the study of the influence of the celestial reference frame, we considered the ICRF2 and ICRF3 compatible solutions of five IVS ACs, finding that the median values of the differences in the corrections computed with each frame were in a band of ±5 μas for all the periods considered except for one of the solutions that presented a more scattered behaviour. These differences were one order of magnitude smaller than the magnitude of the estimated corrections to the nutation terms, which were in the order of a few tens of μas.

Regarding the stochastic model of the least-squares adjustment, we found that using modified uncertainties based on the VLBI network volume brought the smallest a posteriori variance factor with the χ^2^ value closest to one. We concluded that this approach was the most suitable stochastic model to be used in the weighted least-squares adjustment to estimate corrections to the IAU 2000A model, instead of using the original uncertainties or not considering the stochastic model in the least-squares adjustment. We compared the corrections obtained with each stochastic model and we concluded that the choice of the stochastic model could have an effect of a few factors of μas in the estimated corrections, which were significant given that they were in the level of the accuracy of the IAU 2000A nutation model.

## Figures and Tables

**Figure 1 sensors-21-08276-f001:**
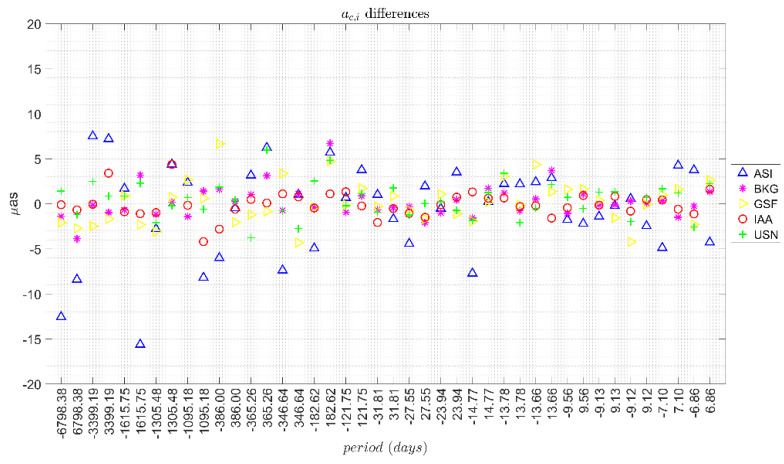
Differences in *a_c,i_* terms derived from the ICRF3 and ICRF2 series.

**Figure 2 sensors-21-08276-f002:**
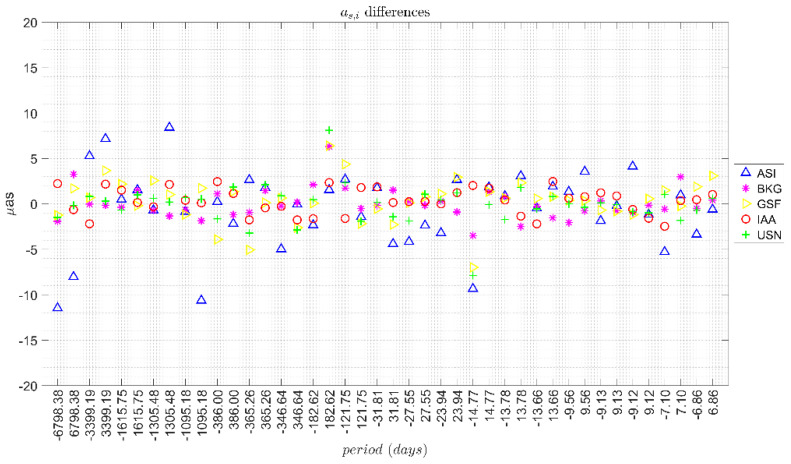
Differences in *a_s,i_* terms derived from the ICRF3 and ICRF2 series.

**Figure 3 sensors-21-08276-f003:**
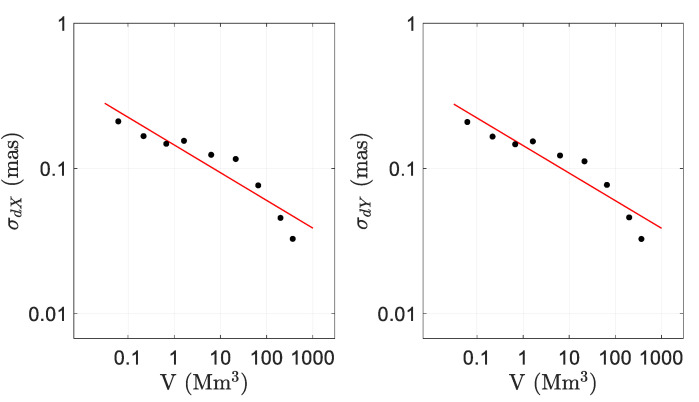
Relationship between the CPO precision and the VLBI network volume.

**Table 1 sensors-21-08276-t001:** Harmonic terms of the nutation model considered in the analysis.

i	*l*	*l’*	*F*	*D*	Ω	Period (Days)
1	0	0	0	0	1	−6798.38
2	0	0	0	0	−1	6798.38
3	0	0	0	0	2	−3399.19
4	0	0	0	0	−2	3399.19
5	2	0	−2	0	−2	−1615.75
6	−2	0	2	0	2	1615.75
7	2	0	−2	0	−1	−1305.48
8	−2	0	2	0	1	1305.48
9	2	0	−2	0	0	−1095.18
10	−2	0	2	0	0	1095.18
11	0	−1	0	0	−1	−386.00
12	0	1	0	0	1	386.00
13	0	−1	0	0	0	−365.26
14	0	1	0	0	0	365.26
15	0	−1	0	0	1	−346.64
16	0	1	0	0	−1	346.64
17	0	0	−2	2	−2	−182.62
18	0	0	2	−2	2	182.62
19	0	−1	−2	2	−2	−121.75
20	0	1	2	−2	2	121.75
21	1	0	0	−2	0	−31.81
22	−1	0	0	2	0	31.81
23	−1	0	0	0	0	−27.55
24	1	0	0	0	0	27.55
25	−1	0	−2	2	−2	−23.94
26	1	0	2	−2	2	23.94
27	0	0	0	−2	0	−14.77
28	0	0	0	2	0	14.77
29	−2	0	0	0	0	−13.78
30	2	0	0	0	0	13.78
31	0	0	−2	0	−2	−13.66
32	0	0	2	0	2	13.66
33	1	0	−2	−2	−2	−9.56
34	−1	0	2	2	2	9.56
35	−1	0	−2	0	−2	−9.13
36	1	0	2	0	2	9.13
37	−1	0	−2	0	−1	−9.12
38	1	0	2	0	1	9.12
39	0	0	−2	−2	−2	−7.10
40	0	0	2	2	2	7.10
41	−2	0	−2	0	−2	−6.86
42	2	0	2	0	2	6.86

**Table 2 sensors-21-08276-t002:** List of VLBI global solutions considered in Test A ^1^.

AC	ICRF2-Compatible Series	ICRF3-Compatible Series
ASI	asi2018a.eops	asi2020a.eops
BKG	bkg00014.eoxy	bkg2020a.eoxy
GSF	gsf2016a.eoxy	gsf2020a.eoxy
IAA	iaa2017a.eops	iaa2021a.eops
USN	usn2018b.eoxy	usn2021b.eoxy

Note: ^1^ Available at ftp://ivs.bkg.bund.de/pub/vlbi/ivsproducts/eops/ (accessed on 6 December 2021).

**Table 3 sensors-21-08276-t003:** Test A: statistics of the corrections to the main lunisolar nutation terms (units: μas).

AC	Ranges (Mean Formal Error)				ICRF3—ICRF2 Differences			
	*a_c,i_*		*a_s,i_*		*a_c,i_*		*a_s,i_*	
	ICRF2	ICRF3	ICRF2	ICRF3	Median	STD	Median	STD
ASI	28.4 (2.7)	26.1 (2.2)	36.5 (2.7)	28.7 (2.2)	0.4	5.2	−0.1	4.3
BKG	22.3 (2.3)	24.7 (2.2)	26.7 (2.3)	28.0 (2.2)	−0.1	1.8	−0.2	1.7
GSF	23.0 (2.1)	26.6 (1.9)	22.6 (2.1)	27.7 (1.9)	−0.1	2.4	0.6	2.4
IAA	21.1 (2.2)	22.6 (2.2)	21.4 (2.2)	22.8 (2.2)	−0.2	1.4	0.4	1.4
USN	21.9 (2.1)	28.0 (1.9)	27.1 (2.1)	31.1 (1.9)	0.7	2.0	0.4	1.4
Median	22.3 (2.2)	26.1 (2.2)	26.7 (2.2)	28.0 (2.2)	−0.1	2.0	0.4	1.7

**Table 4 sensors-21-08276-t004:** Average network volume, number of sessions and CPO uncertainties per volume group.

*V* Group	Number of Sessions	$\bar{V}$(Mm^3^)	$\bar{\sigma_{dX}}\;(\mathrm{mas})$	$\bar{\sigma_{dY}}\;(\mathrm{mas})$
*V* < 0.1	13	0.062	0.187	0.187
0.1 ≤ *V* < 0.316	21	0.221	0.135	0.135
0.316 ≤ *V* < 1	62	0.656	0.127	0.127
1 ≤ *V* < 3.16	68	1.854	0.129	0.125
3.16 ≤ *V* < 10	114	6.244	0.096	0.098
10 ≤ *V* < 31.6	198	21.655	0.091	0.091
31.6 ≤ *V* < 100	503	64.781	0.068	0.069
100 ≤ *V* < 316	1055	180.343	0.049	0.049
316 ≤ *V* < 1000	234	377.666	0.045	0.045

**Table 5 sensors-21-08276-t005:** Estimated coefficients of the power model relating precision and network volume.

EOP	*b* (mas)	*c* (mas)
*dX*	−2.097	−0.154
*dY*	−2.096	−0.154

**Table 6 sensors-21-08276-t006:** Statistical indicators of the solutions obtained using different stochastic models for the fitting of corrections to the IAU 2000A model.

Test	$\hat{\sigma }$ (μas)	χ^2^	Mean Formal Error (μas)	
			*a_c,i_*	*a_s,i_*
Test B.1: no stochastic model	119.0	2.1	1.8	1.8
Test B.2: original uncertainties	169.1	4.2	1.9	1.9
Test B.3: modified uncertainties	104.1	1.6	1.4	1.4

**Table 7 sensors-21-08276-t007:** Test B: statistics of the differences in the corrections (units: μas).

	*a_c,i_*			*a_s.i_*		
	Median	STD	Range	Median	STD	Range
Test B.1–Test B.3	−0.2	4.5	20.1	−0.9	4.3	14.1
Test B.2–Test B.3	−1.0	4.5	19.6	−0.9	4.9	17.8

## Data Availability

The datasets analysed in this study are freely available at the IVS servers.

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
