# Peer review of "The Celestial Frame and the Weighting of the Celestial Pole Offsets in the Computation of VLBI-Based Corrections for the Main Lunisolar Nutation Terms"

_sensors, 2021, doi:10.3390/s21248276_

Round 1
Reviewer 1 Report
Dear authors,
Please find my review report in the attached pdf file.
Best regards,

Author Response
Please find our notes attached

Reviewer 2 Report
The author studied the influence of CPO on choice of ICRF, and weighting strategies. While, their conclusions should be improved. To give some exact statement, not just difference. When fitting a_c_i and a_s_i, there are 42 terms, then there should be 42 pair of (a_c_i, a_s_i) ? Why in Figure 1 and Figure 2, they only show one a_c_i and a_s_i ? For fitting, the residual or chi_squares should be given, so that one can make a judgement, which fitting is better, for example, to yield some conclusions, such as ICRF3 better than ICRF2 or IAA better or worse than USN ? But there is no such kind of information. When they study the influcence of ICRF, is it possible to evaluate whether ICRF3 is better or not than ICRF2 ? I guess some reader would be interested with this question. Since, on average, ICRF3 should be a little bit more accurate than ICRF2. For their second points, the conclusion is also ambiguous. For me, from Table 5, it is very clear that Test B.3: modified uncertainties is better than Test B.1 and Test B.2. However, in Table 6, the conclusions looks conflict with that of Table 5, as B.1 and B.2 looks more consistent. Thus one should make a choice, which strategy is better ? Since Table 6 only means B.1 and B.2 are more similar, but not mean they are better. Similar does not mean better. As In Table 5 B.3 shows a lower of chi_squre, this is a clear conclusion that B.3 strategy is better than B.1 and B.2. Thus, maybe, its better to just stop at table 5, and made a exact statement that B.3 strategy is statistically better than B.1 and B.2. There are some minor type errors, for example, (1) page 4, line 85, the subscript of a_c should be a_c,i. (2) Figure 3, the xticks values seems be wrong.Author Response
Please find our notes attached

Round 2
Reviewer 2 Report
The author investigate the influence of weighting strategy and versions of celestial frame on the corrections for the lunisolar nutation terms. They found no significant dependence on the choice of ICRF, but did find the modified weighting strategy can yields lowerer chi_square, which indicating such weighting strategy is statistically better than other two weighting strategy. The conlcusion is reliable and useful for forthcoming studies.